# Non-Inversion Tillage as a Chance to Increase the Biodiversity of Ground-Dwelling Spiders in Agroecosystems: Preliminary Results

Elżbieta Topa [1], Agnieszka Kosewska [1], Mariusz Nietupski [1,*], Łukasz Trębicki [2], Łukasz Nicewicz [3] and Izabela Hajdamowicz [4]

[1] Department of Entomology, Phytopathology and Molecular Diagnostics, University of Warmia and Mazury in Olsztyn, Prawochenskiego 17, 10-719 Olsztyn, Poland; topa@uwm.edu.pl (E.T.); a.kosewska@uwm.edu.pl (A.K.)

[2] Department of Invertebrate Zoology and Hydrobiology, University of Lodz, Banacha 12/16, 90-237 Lodz, Poland; lukasz.trebicki@biol.uni.lodz.pl

[3] Department of Animal Physiology and Ecotoxicology, University of Silesia in Katowice, Bankowa 9, 40-007 Katowice, Poland; lnicewicz@us.edu.pl

[4] Flächenagentur B-W, Gerhard-Koch-Straße 2, 73760 Ostfildern, Germany; hajdamo@gmail.com

* Correspondence: mariusz.nietupski@uwm.edu.pl

**Abstract:** Spiders (Araneae) create abundant and diverse assemblages in many agroecosystems, where they play a crucial role as the main group of predators and pest controllers. However, seasonal disturbance in the agricultural environment (e.g., harvesting or ploughing) affects spider assemblages. The main aim of this research was to compare assemblages of Araneae colonising cereal fields cultivated under two different systems of soil tillage: conventional with ploughing and non-inversion tillage. The research covered plantations of triticale, wheat, and barley, situated in northeastern Poland. Ground-dwelling spiders were captured into modified pitfall traps filled up to 1/3 height with an ethylene glycol solution. In total, 6744 spiders representing 67 species classified in 13 families were caught. The traps were emptied every two weeks from the end of April until the end of July. A total of 2410 specimens representing 55 species were captured in the fields with simplified cultivation, while the remaining 4334 specimens representing 49 species were trapped in conventional fields where ploughing was performed. The Shannon diversity (H') and evenness (J') indices reached higher values in the fields without ploughing. According to IndVal *Erigone, dentipalpis* and *Bathyphantes gracilis* were signifi-cantly characteristic (*p* < 0.05) for non-inversion soil tillage, whereas six species, *Oedothorax apicatus, Pardosa prativaga, Pardosa paludicola, Pachygnatha clerki, Dicimbium nigrum brevisetosum,* and *Clubiona reclusa,* were typical of soil tillage with ploughing. The research showed that simplification of soil tillage in cereal fields improves the biodiversity of arachnofauna in agricultural ecosystems. The use of conventional tillage systems with ploughing promotes agrobiontic species of the families Linyphiidae and Lycosidae.

**Keywords:** ground-dwelling spiders; soil tillage systems; cereal crops; biodiversity

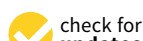

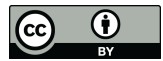

## 1. Introduction

Concepts of integrated pest management (IPM) have been recommended in agricultural production for years and are mandatory according to EU Directive (2009/128/EC), with a preference for non-chemical methods, among which the agrotechnical and biological methods respond well to the assumptions underlying the current plant protection knowledge and practice. In line with contemporary systems of soil tillage and plant cultivation, the goals are to produce high-quality food and to preserve the environmental values of agroecosystems and the agricultural landscape [1]. In recent years, we have witnessed the development of new forms of agrotechnical soil tillage. As the conventional ploughing system is expensive, various systems of ploughless soil tillage, which have a beneficial

impact on the environment, are increasingly implemented [2–4]. Conservation tillage is predominantly ploughless cultivation and typically does not displace soil-dwelling organisms [4]. Therefore, spiders and other soil-dwelling invertebrates potentially have a greater chance of survival [5–7]. Naturally existing generalist predators, such as spiders or ground living beetles, contribute to the important ecosystem service of the biological control of insect crop pests [8]. Spiders are among the most abundant predators in many agroecosystems and are the most diverse generalist predators [9–11]. What makes spiders different from other predatory invertebrates is the variety of hunting strategies, and we can distinguish ground hunters, foliage runners, and web-building spiders [12–14]. Crop fields in Europe are distinguished by the prevalence of agrobiontic spiders of the families Linyphiidae and Lycosidae. These species have relatively short life cycles and high dispersion abilities, which allow them to colonise new fields every year and to produce offspring before harvest or ploughing [15–18]. Generally, the lack of foraging specialisation among most spiders is considered to be an important factor that limits the occurrence of many pests on cropped fields, especially aphids. Such species as the bird cherry-oat aphid *Rhopalosiphum padi* (L.), rose grain aphid *Metopolophium dirhodum* (Walker), and grain aphid *Sitobion avenae* Fabricius, which are important vectors of viruses, belong to the set of prey hunted by many spiders, both ground hunters and web weavers [13,19–22]. Flies are another group of insects that make up a large share of the spider diet. The following are often mentioned: frit fly (*Oscinella frit* L.), wheat midge (*Sitodiplosis mosellana* Gehin), hessian fly (*Mayetiola destructor* Say), and wheat fly (*Contarinia tritici* Kirby) [13]. Maintaining the biodiversity of spiders in agrocenoses plays an important role in keeping the number of pest insects below the harmfulness thresholds [13,23], especially in the early phase of the plant growing season, when the number of individuals in assemblages of predatory insects is insufficient [24,25]. Simplified soil tillage systems such as direct sowing, only harrowing, ploughing once in a few years, or non-inversion soil tillage have a positive influence on the stability of the environment, raising the species diversity of ground predators [2,26], whereas the effect of ploughing soil tillage systems on Araneae is equivocal. Schmidt et al. [21] point out the high sensitivity of spiders to a ploughing soil tillage system. On the other hand, Duffey [27] claimed that ploughing did not have any effect on spiders from the Linyphiidae family.

The main aim of the study was to compare assemblages of Araneae colonising cereal fields cultivated under two different systems of soil tillage: conventional with plough (pl) and non-inversion (ni). The following research hypotheses were made:

- Ploughing causes a decrease in the abundance, species richness, and biodiversity of spiders in the growing season;
- In fields under the non-inversion tillage system, which are less disturbed, there are more active epigeic hunters (cursorials), whereas web spiders are more abundant in conventional fields.

## 2. Materials and Methods

### 2.1. Study Area

The study was carried out in Tomaszkowo, a village in the northeastern part of Poland (GPS 53,718725; 20,414701), in 2016. The observations were carried out on farms with fields cropped with winter cereals in the conventional plough system (furrow slice turning ploughs were used, followed by a tiller harrow to prepare the soil for sowing) and in the ploughless system of soil tillage (a special soil mixing cultivation aggregates were used without turning furrows or slicing). At the end of summer 2015, agrotechnical procedures related to the preparation of soil for sowing were carried out in the studied fields. After sowing, plant protection and fertilisation treatments were carried out (Table 1).

**Table 1.** Characteristic of cereals plantations alongside the specification of pesticides applied and fertilisation.

| | | Combination | | | |
|---|---|---|---|---|---|
| | | T pl | W pl | T ni | B ni |
| | Cereal Cultivar DS [1] Area | Triticale Baltico 7 September 2015 38 ha | Wheat Bamberka 15 September 2015 21.5 ha | Triticale Elpaso 10 September 2015 60 ha | Barley Titus 17 September 2015 22 ha |
| Herbicide | AS [2] | metribuzin + flufenacet + chlorsulfuron | metribuzin + flufenacet + chlorsulfuron carfentrazone -ethyl | chlorsulfuron + metribuzin + isoproturon | chlorsulfuron |
| | DA [3] | 14 September 2015 | 21 September 2015 | 5 October 2015 | 24 September 2015 |
| | AS | pinoxaden | carfentrazone-ethyl | methyl sodium iodosulfuron + methyl mesosulfuron | pinoxaden |
| | DA | 26 April 2016 | 30 March 2016 | 25 April 2016 | 25 April 2016 |
| Fungicide | AS | propiconazole + fenpropidin | propiconazole + fenpropidin | prothioconazole + spiroxamine | prothioconazole + spiroxamine |
| | DA | 30 April 2016 | 30 April 2016 | 16 May 2016 | 16 May 2016 |
| | AS | tebuconazole | epoxiconazole | fluoxastrobin + prothioconazole | fluoxastrobin + prothioconazole |
| | DA | 2 June 2016 | 27 May 2016 | 15 June 2016 | 15 June 2016 |
| | AS | | tebuconazole | | |
| | DA | | 26 June 2016 | | |
| Insecticide | AS | deltamethrin | deltamethrin | deltamethrin | deltamethrin |
| | DA | 15 June 2016 | 15 June 2016 | 15 June 2016 | 15 June 2016 |
| NPK fertilisation | DA | 3 March, 12 April, 18 May and 2 June 2016 | 29 March, 12 April and 27 May 2016 | 17 September 2015, 15 April, 28 April 2016 | 9 October 2015, 15 April, 28 April 2016 |

[1] DS—date of sowing; [2] AS—active substance; [3] DA—date of application.

### 2.2. Data Collection

A method of modified pitfall traps (plastic cups 10 cm diameter, 15 cm deep), filled to 1/3 height with a solution of ethyl glycol, was used to capture arachnofauna. Two plots were established in each of the four selected fields in two soil tillage systems. The distance between the plots was about 300 m. On each plot, 5 traps were set up in line at a 10-m distance from each other. Thus, there were 20 traps per treatment (2 fields × 2 plots × 5 traps). The first trap on each plot was placed about 50 m from the field edge. The traps were emptied every two weeks, from the end of April to the end of July. In total, it makes eight collection dates when traps were emptied.

Observations of epigeic spiders were carried out with the help of a quantitative method, taking into account the number of specimens, number of species, and families. The identified material [28] was classified into different ecological groups according to Hängii et al. [12], Uetz et al. [29], and Birkhofer et al. [10]. Spiders were divided into two groups in relation to their hunting strategies. The two groups were spiders building webs (web spiders) and actively hunting ones (cursorial spiders). In addition, agrobiontic species, rapidly colonising agrocenoses, were distinguished [10]. The Górny and Grüm scale of dominance was used to determine classes of dominance [30]. The following were considered: superdominant (>30%), eudominant (10–30%), dominant (5–10%), subdominant (2–5%), recedent (1–2%), and subrecedent (<1%).

### 2.3. Data Analysis

The Shannon species diversity index (log base 2.718) (H′) and Pielou evenness index (J′) were used to process the results. Due to a lack of normal distribution, differences in means of parameters describing assemblages (species abundance and richness and ecological groups abundance) were tested using a Poisson generalised linear model (GLM),

which included two factors (the soil cultivation system and the sampling period). Indirect ordination of spider assemblages captured in the study area was performed using non-metric multidimensional scaling (NMDS), in which the Bray–Curtis index was used as an indicator of similarity, and statistical significance was confirmed by analysis of variance of similarity (ANOSIM). The obtained graph of NMDS presents data for a two-dimensional solution. Redundancy analysis (RDA) was used to investigate correlations between ecological groups of spiders (agrobionts, cursorial, web spiders) and two environmental variables: type of tillage and cereal. The RDA method was chosen following the DCA data distribution analysis, which was linear. Data were non-transformed. The IndVal (indicator species analysis) method [31] with the "labdsv" package using R software [32] was used to identify indicator species in each field. This method enables the user to find indicator species and species assemblages characterising groups of sites. The IndVal value is the result of the specificity and fidelity measure. The maximum value (100) of this indicator is reached when all individuals of a species are found in a single group of sites (high specificity) and when the species occurs in all sites of that group (high fidelity). The statistical significance of indicator values was estimated using the Monte Carlo permutations test. All statistical calculations and their graphic interpretation were performed using the following software: Statistica 13.1 (Dell Inc. Tulusa, OK, USA), Canoco 4.51 (Biometris–Plant Research International, Wageningen, The Netherlands), and R software (The R Foundation for Statistical Computing).

## 3. Results

A total of 6744 ground-dwelling spiders were collected, representing 67 species in 13 families (Table 2). More specifically, 2410 specimens representing 55 species were captured in the fields with simplified cultivation (ni) while the remaining 4334 individuals belonging to 49 species were trapped in conventional fields (pl) where the soil was ploughed. The Shannon diversity (H′) and Pielou evenness (J′) indices reached higher values in the fields without ploughing (Table 2).

**Table 2.** Species composition, abundance, and indices describing ground-dwelling spiders occurring in cereals (T-triticale, B-barley, W-wheat) cultivated in ploughing (pl) and non-inversion (ni) tillage system.

| Family | Species | Ecological Description [1] | Cereal Soil Tillage | | | |
|---|---|---|---|---|---|---|
| | | | **T ni** | **B ni** | **T pl** | **W pl** |
| Linyphiidae | *Oedothorax apicatus* (Blackwall, 1850) | A, C | 553 | 511 | 1711 | 975 |
| Linyphidae | *Erigone atra* (Blackwall, 1833) | A, W | 178 | 130 | 293 | 38 |
| Linyphidae | *Erigone dentipalpis* (Wider, 1834) | A, W | 121 | 182 | 216 | 15 |
| Lycosidae | *Pardosa prativaga* (L. Koch, 1870) | A, C | 98 | 89 | 90 | 220 |
| Lycosidae | *Pardosa paludicola* (Clerck, 1758) | C | 8 | 47 | 33 | 188 |
| Lycosidae | *Pardosa palustris* (Linnaeus, 1758) | A, C | 32 | 108 | 44 | 89 |
| Lycosidae | *Trochosa ruricola* (De Geer, 1778) | A, C | 21 | 24 | 9 | 51 |
| Thomisidae | *Xysticus kochi* (Thorell, 1872) | A, C | 6 | 30 | 3 | 42 |
| Linyphiidae | *Bathyphantes gracilis* (Blackwall, 1841) | A, W | 36 | 26 | 15 | 3 |
| Tetragnathidae | *Pachygnatha degeeri* (Sundevall, 1830) | A, C | 11 | 26 | 27 | 15 |
| Lycosidae | *Pardosa agrestis* (Westring, 1861) | A, C | 7 | 16 | 13 | 21 |
| Lycosidae | *Pardosa pullata* (Clerck, 1758) | A, C | 9 | 17 | 17 | 12 |
| Linyphiidae | *Oedothorax retusus* (Westring, 1851) | A, C | 10 | 0 | 17 | 27 |
| Tetragnathidae | *Pachygnatha clercki* (Sundevall, 1823) | A, C | 2 | 1 | 14 | 9 |
| Tetragnathidae | *Tetragnatha extensa* (Linnaeus, 1758) | A, W | 5 | 3 | 18 | 0 |
| Linyphiidae | *Tenuiphantes tenebricola* (Wider, 1834) | W | 0 | 11 | 0 | 6 |

**Table 2.** *Cont.*

| Family | Species | Ecological Description [1] | Cereal Soil Tillage | | | |
|---|---|---|---|---|---|---|
| | | | T ni | B ni | T pl | W pl |
| Linyphiidae | *Araeoncus humilis* (Blackwall, 1841) | A, W | 3 | 4 | 4 | 2 |
| Linyphiidae | *Dicymbium nigrum brevisetosum* (Locket, 1962) | W | 1 | 0 | 12 | 0 |
| Lycosidae | *Pardosa amentata* (Clerck, 1757) | A, C | 0 | 7 | 0 | 6 |
| Lycosidae | *Pardosa lugubris* (Walckenaer, 1802) | C | 1 | 6 | 1 | 4 |
| Thomisidae | *Xysticus cristatus* (Clerck, 1758) | A, C | 1 | 5 | 2 | 2 |
| Lycosidae | *Pardosa riparia* (C.L. Koch, 1833) | C | 2 | 4 | 2 | 1 |
| Pisauridae | *Pisaura mirabilis* (Clerck, 1758) | C | 1 | 1 | 2 | 5 |
| Lycosidae | *Alopecosa cuneata* (Clerck, 1758) | C | 0 | 5 | 1 | 2 |
| Clubionidae | *Clubiona reclusa* (O.P.-Cambridge, 1863) | C | 0 | 0 | 1 | 7 |
| Thomisidae | *Ozyptila trux* (Blackwall, 1846) | C | 1 | 3 | 3 | 1 |
| Lycosidae | *Trochosa terricola* (Thorell, 1856) | C | 1 | 0 | 1 | 6 |
| Linyphiidae | *Neriene clathrata* (Sundevall, 1830) | W | 0 | 4 | 0 | 3 |
| Linyphiidae | *Stemonyphantes lineatus* (Linnaeus, 1758) | W | 1 | 0 | 0 | 6 |
| Linyphidae | *Diplostyla concolor* (Wider, 1834) | A, W | 1 | 1 | 2 | 2 |
| Linyphiidae | *Savignia frontata* (Blackwall, 1833) | W | 1 | 1 | 3 | 0 |
| Miturgidae | *Zora spinimana* (Sundevall, 1833) | C | 1 | 2 | 0 | 1 |
| Hahniidae | *Hahnia pusilla* (C.L. Koch, 1841) | W | 0 | 1 | 0 | 2 |
| Linyphiidae | *Porrhomma pygmaeum* (Blackwall, 1834) | W | 1 | 1 | 1 | 0 |
| Thomisidae | *Xysticus ulmi* (Hahn, 1831) | A, C | 1 | 1 | 0 | 1 |
| Gnaphosidae | *Zelotes subterraneus* (C.L. Koch, 1833) | C | 1 | 2 | 0 | 0 |
| Linyphiidae | *Agyneta affinis* (Kulczyński, 1898) | A, W | 2 | 0 | 0 | 0 |
| Lycosidae | *Alopecosa pulverulenta* (Clerck, 1758) | A, C | 1 | 1 | 0 | 0 |
| Linyphiidae | *Bathyphantes parvulus* (Westring, 1851) | W | 1 | 0 | 0 | 1 |
| Linyphiidae | *Centromerus sylvaticus* (Blackwall, 1841) | W | 1 | 0 | 1 | 0 |
| Clubionidae | *Clubiona subtilis* (L. Koch, 1867) | C | 2 | 0 | 0 | 0 |
| Linyphidae | *Erigonella hiemalis* (Blackwall, 1841) | W | 0 | 2 | 0 | 0 |
| Araneidae | *Larinioides patagiatus* (Clerck, 1758) | W | 0 | 2 | 0 | 0 |
| Linyphiidae | *Pocadicnemis juncea* (Locket and Millidge, 1953) | W | 1 | 1 | 0 | 0 |
| Theridiidae | *Robertus lividus* (Blackwall, 1836) | W | 1 | 0 | 1 | 0 |
| Lycosidae | *Trochosa spinipalpis* (F.P.-Cambridge, 1895) | C | 0 | 0 | 1 | 1 |
| Linyphiidae | *Walckenaeria antica* (Wider, 1834) | W | 0 | 0 | 2 | 0 |
| Linyphiidae | *Walckenaeria nudipalpis* (Westring, 1851) | W | 0 | 0 | 2 | 0 |
| Linyphiidae | *Agyneta rurestris* (C. L. Koch, 1836) | A, W | 0 | 1 | 0 | 0 |
| Araneidae | *Argiope bruennichi* (Scopoli, 1772) | W | 0 | 0 | 0 | 1 |
| Linyphiidae | *Ceratinella brevis* (Wider, 1834) | W | 0 | 1 | 0 | 0 |
| Clubionidae | *Clubiona neglecta* (O.P.-Cambridge, 1862) | C | 0 | 1 | 0 | 0 |
| Gnaphosidae | *Drassyllus lutetianus* (L. Koch, 1866) | C | 0 | 0 | 0 | 1 |
| Mimetidae | *Ero furcata* (Villers, 1789) | W | 0 | 1 | 0 | 0 |
| Linyphiidae | *Kaestneria pullata* (O.P.-Cambridge, 1863) | W | 1 | 0 | 0 | 0 |
| Araneidae | *Mangora acalypha* (Walckenaer, 1802) | W | 0 | 1 | 0 | 0 |

**Table 2.** *Cont.*

| Family | Species | Ecological Description [1] | Cereal Soil Tillage | | | |
|---|---|---|---|---|---|---|
| | | | T ni | B ni | T pl | W pl |
| Araneidae | *Neoscona adianta* (Walckenaer, 1802) | W | 0 | 1 | 0 | 0 |
| Linyphiidae | *Oedothorax fuscus* (Blackwall, 1841) | A, C | 0 | 1 | 0 | 0 |
| Linyphiidae | *Oedothorax gibbosus* (Blackwall, 1841) | C | 0 | 0 | 0 | 1 |
| Thomisidae | *Ozyptila praticola* (C.L. Koch, 1837) | C | 0 | 1 | 0 | 0 |
| Lycosidae | *Piratula uliginosa* (Thorell, 1856) | C | 0 | 0 | 0 | 1 |
| Linyphiidae | *Tallusia experta* (O.P.-Cambridge, 1871) | W | 1 | 0 | 0 | 0 |
| Linyphiidae | *Tenuiphantes cristatus* (Menge, 1866) | W | 1 | 0 | 0 | 0 |
| Philodromidae | *Thanatus striatus* (C.L. Koch, 1845) | C | 0 | 0 | 1 | 0 |
| Philodromidae | *Tibellus maritimus* (Menge, 1874) | C | 0 | 0 | 1 | 0 |
| Philodromidae | *Tibellus oblongus* (Walckenaer, 1802) | C | 0 | 0 | 1 | 0 |
| Gnaphosidae | *Zelotes petrensis* (C.L. Koch, 1839) | C | 0 | 0 | 0 | 1 |
| Total Individuals | | | 1127 | 1283 | 2565 | 1769 |
| | | | 2410 | | 4334 | |
| Total Species | | | 40 | 43 | 36 | 38 |
| | | | 55 | | 49 | |
| Shannon H′ Log Base 2.718 | | | 1.814 | 2.177 | 1.35 | 1.744 |
| | | | 2.052 | | 1.613 | |
| Shannon J′ | | | 0.492 | 0.579 | 0.377 | 0.479 |
| | | | 0.512 | | 0.415 | |

[1] A-agrobionts, W-web spiders, C-cursorial.

The abundance of spiders and species diversity (Shannon H′) depended significantly on field tillage and season (Table 3).

**Table 3.** Results of the GLM test of significance (Wald statistic) of the effect of type of soil tillage and period when sampled on abundance and species richness of spiders and abundance of ecological groups.

| | df | Wald Stat. | $p$ |
|---|---|---|---|
| Abundance | | | |
| Date | 7 | 1728.20 | 0.00 |
| Tillage | 1 | 266.73 | 0.00 |
| Date × Tillage | 7 | 357.00 | 0.00 |
| Richness | | | |
| Date | 7 | 240.19 | 0.00 |
| Tillage | 1 | 2.35 | 0.13 |
| Date × Tillage | 7 | 36.22 | 0.00 |
| Shannon H′ | | | |
| Date | 7 | 49.12 | 0.00 |
| Tillage | 1 | 6.36 | 0.01 |
| Date × Tillage | 7 | 1.80 | 0.97 |
| Agrobionts | | | |

**Table 3.** *Cont.*

|  | df | Wald Stat. | p |
|---|---|---|---|
| Date | 7 | 1642.65 | 0.00 |
| Tillage | 1 | 244.14 | 0.00 |
| Date × Tillage | 7 | 327.54 | 0.00 |
| Web spiders |  |  |  |
| Date | 7 | 379.26 | 0.00 |
| Tillage | 1 | 0.60 | 0.44 |
| Date × Tillage | 7 | 103.30 | 0.00 |
| Cursorial |  |  |  |
| Date | 7 | 1402.70 | 0.00 |
| Tillage | 1 | 388.50 | 0.00 |
| Date × Tillage | 7 | 242.51 | 0.00 |

The abundance of ground-dwelling spiders in the plough tillage system was significantly higher throughout almost the entire research period, except the last 10 days of June (Figure 1a). A reverse dependence was noted for the Shannon species diversity index (H′), which was higher in the ploughless system during the whole research period (Figure 1b).

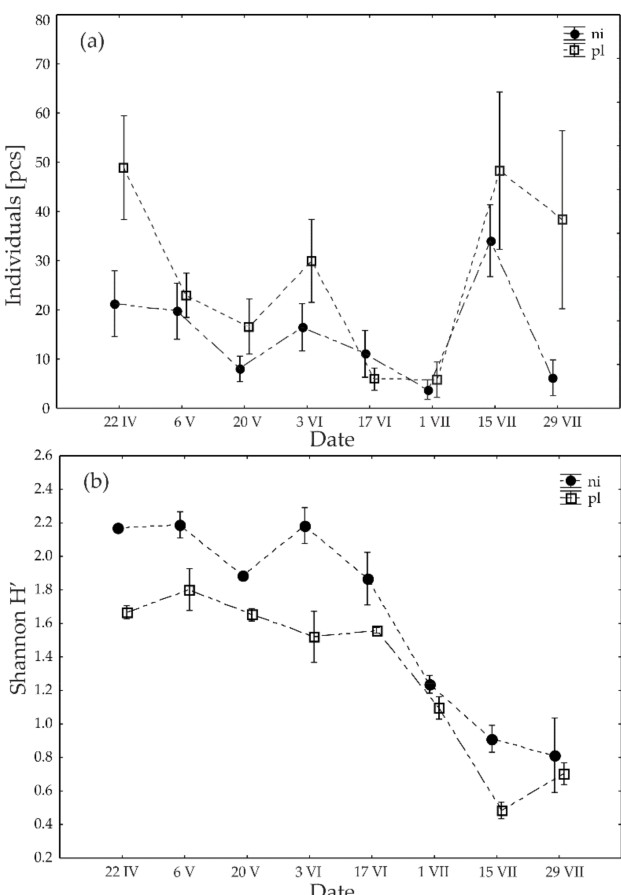

**Figure 1.** *Cont.*

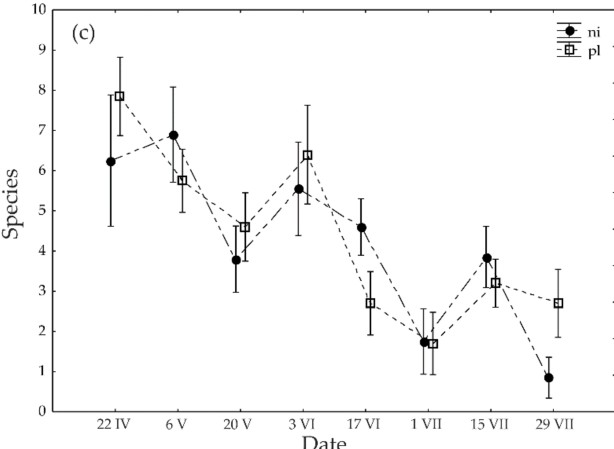

**Figure 1.** Average abundance (**a**), species diversity (**b**), and species richness (**c**) of spiders depended on studied soil tillage (ni-non-inversion and pl-ploughed) in cereals fields (the vertical lines indicate 0.95 confidence interval).

However, in both soil tillage systems, a rapid decrease in the value of this index was noted at the beginning of July. There were no significant differences between fields with different soil tillage in terms of spider richness, although the effect was also dependent on the seasonal dynamics (Table 3). A decrease in the number of individuals captured in ploughed fields at the end of June coincided with a significant decrease in the number of captured species (Figure 1c).

The non-metric multidimensional scaling (NMDS) diagram displays similarities and differences in the species composition and abundance of ground-dwelling spiders in fields cropped with the analysed cereals and maintained in two soil tillage systems (Figure 2). The ANOSIM analysis confirmed significant differences between the analysed assemblages of spiders (R: 0.6; *p*: 0.0001). The NMDS diagram revealed that spiders assemblage from ploughed fields of wheat was significantly different from the other assemblages. Fields without ploughing, irrespective of the cereal grown (triticale or barley), showed close similarity between the spider assemblages that had settled there. The most numerous species living in the studied fields was *Oedothorax apicatus*, which, regardless of the soil tillage system applied, was invariably in the group of superdominant species and composed 44.15% of spider assemblages in the ploughless fields (ni) and 61.98% of spider assemblages in fields with the conventional soil tillage system (pl) (Table 4).

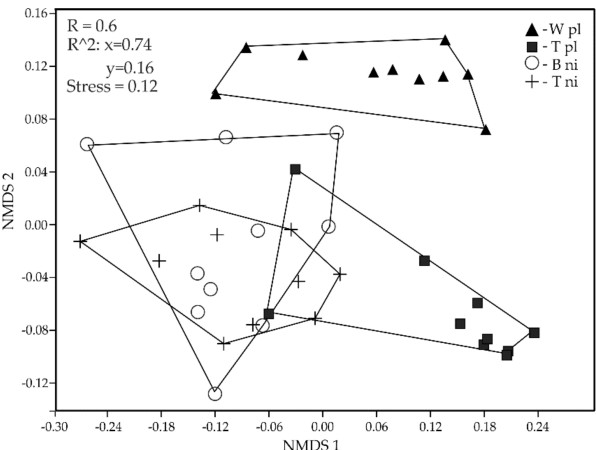

**Figure 2.** Diagram of non-metric multidimensional scaling (NMDS) performed using the Bray-Curtis similarity matrix of spiders assemblages (B ni-non-inversion barley; T pl-ploughed triticale, W pl-ploughed wheat, T ni-non-inversion triticale).

**Table 4.** The most abundant spider species collected in two tillage systems. The Górny and Grüm scale of dominance was used [30].

| Dominance Class | |
| --- | --- |
| **Non-Inversion Tillage** | **Ploughing Tillage** |
| Superdominant species (>30%) | |
| *Oedothorax apicatus* (44.15%) | *Oedothorax apicatus* (61.98%) |
| Eudominant species (10–30%) | |
| *Erigone atra* (12.78%), *Erigone dentipalpis* (12.57%) | |
| Dominant species (5–10%) | |
| *Pardosa prativaga* (7.76%), *Pardosa palustris* (5.81%) | *Erigone atra* (7.64%), *Pardosa prativaga* (7.15%), *Erigone dentipalpis* (5.33%), *Pardosa paludicola* (5.1%) |
| Subdominant species (2–5%) | |
| *Bathyphantes gracilis* (2.57%), *Pardosa paludicola* (2.28%) | *Pardosa palustris* (3.07%) |
| Recedent species (1–2%) | |
| *Trochosa ruricola* (1.87%), *Pachygnatha degeeri* (1.54%), *Xysticus kochi* (1.49%), *Pardosa pullata* (1.08%) | *Trochosa ruricola* (1.38%), *Xysticus kochi* (1.04%), *Oedothorax retusus* (1.02%) |
| Subrecedent species (<1%) | |
| The remaining 45 species (5.15%) | The remaining 40 species (6.3%) |

In fields without ploughing, other numerous species were *Erigone atra* (12.78%) and *Erigone dentipalpis* (12.57%), which were classified as eudominants, a class that was not distinguished in the ploughed fields. These species in fields where the soil was ploughed were in the class of dominants. As for the number of species, the class of subrecedents was quite numerous, as it was composed of 45 species in ploughless fields and 40 species in ploughed ones.

In terms of their hunting strategies, the analysed spiders were divided into two groups: cursorial and web spiders. Additionally, a group of species characteristic for cropped fields, i.e., agrobionts, was distinguished. As for agrobionts and cursorial species, significant differences were observed in their numbers regarding the seasonal presence of both groups as correlated with the type of soil tillage (Table 3). The numbers of specimens of the typical field species, i.e., agrobionts, were significantly higher in the ploughed fields during almost the whole plant growing season (Figure 3a).

Agrobionts appeared in particularly high numbers in the early phase of the growing season. Similar relationships were observed for epigeic hunters, i.e., cursorial species, which were more abundant in the ploughed fields, and were most active in April and in July (Figure 3b). There were no significant differences between fields with the two soil tillage methods in the number of web spiders, although the effect was dependent on seasonal dynamics (Table 3). During the entire plant growing season, strong fluctuations in the numbers of web-building spiders were noted in both types of soil tillage (Figure 3c). The highest number of specimens of these spiders was recorded at the beginning of the plant growing season.

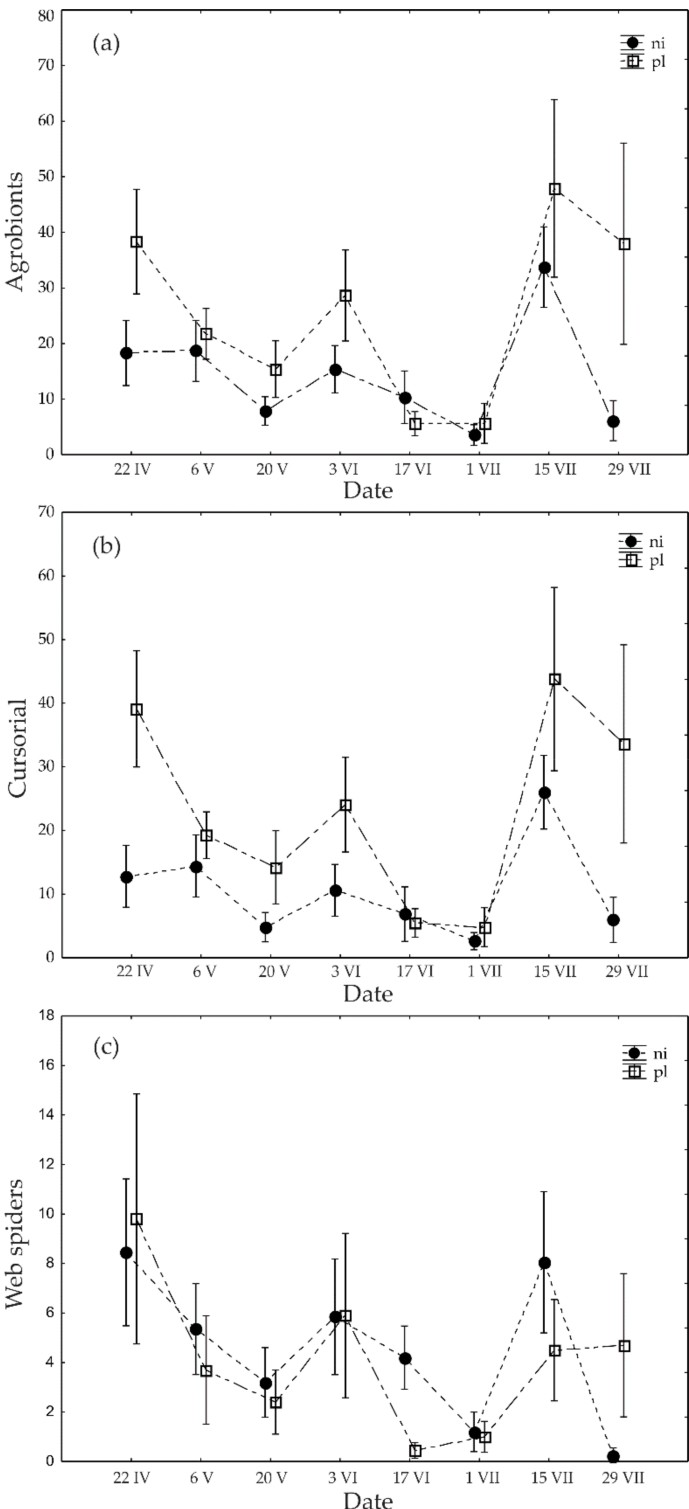

**Figure 3.** Average abundance of agrobionts (**a**), cursorial (**b**), and web spiders (**c**) in relation to studied soil tillage (ni-non-inversion and pl-ploughed) in cereals fields (vertical lines indicate 0.95 confidence interval).

RDA shows dependencies between the analysed ecological groups of spiders and environmental factors, such as the type of soil tillage and the type of cereal grown (Figure 4).

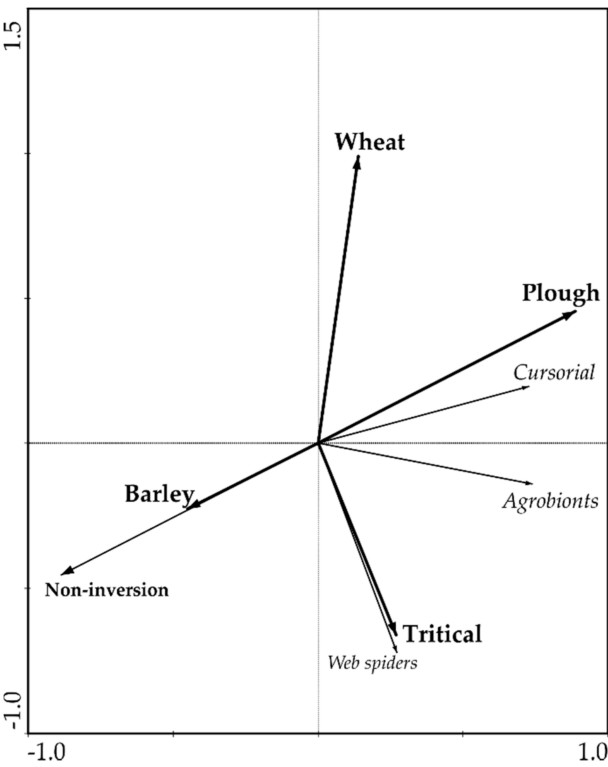

**Figure 4.** Diagram of the RDA demonstrating the relationships between the analysed environmental variables (type of tillage, cereal) and the ecological groups of spiders (agrobionts, cursorial, web).

The first ordination axis, describing 90.8% of the variance, was strongly correlated with the presence of agrobiontic epigeic hunters, i.e., cursorial species, which occur most often in ploughed fields. The presence of web-building spiders was more strongly correlated with the kind of cereal grown, as they most willingly appeared in triticale fields. The IndVal method indicated that two species of spiders, *E. dentipalpis* and *B. gracilis* were characteristic for non-inversion soil tillage, while a set of six species of spiders, with dominant *O. apicatus*, proved to be characteristic for ploughed fields (Table 5).

**Table 5.** Values of IndVal indicator of the spider species significantly characteristic ($p < 0.05$) for the type of soil cultivation in cereal crops.

| | IndVal | *p* | Frequency |
|---|---|---|---|
| Non-Inversion | | | |
| *Erigone dentipalpis* | 32.3 | 0.012 | 150 |
| *Bathyphantes gracilis* | 18.4 | 0.001 | 54 |
| Plough | | | |
| *Oedothorax apicatus* | 60.9 | 0.001 | 258 |
| *Pardosa prativaga* | 30.4 | 0.026 | 137 |
| *Pardosa paludicola* | 21.0 | 0.001 | 62 |
| *Pachygnatha clercki* | 9.4 | 0.001 | 20 |
| *Dicymbium nigrum brevisetosum* | 5.2 | 0.007 | 10 |
| *Clubiona reclusa* | 4.4 | 0.018 | 7 |

## 4. Discussion

For many years, spider assemblages have been a key component of integrated pest management strategies and high-level agents of biological control [8,23,33–35]. Although cereal fields represent habitats that are considerably disturbed every year, they are still characterised by a certain potential for useful fauna, with spiders playing the main role. In this study, we observed an abundant assemblage of spiders, 6744 specimens belonging to 67 species from 13 families. In comparison, other authors Thorbek and Bilde [5], over three years of research on the impact of crop management on generalist arthropod predators from various cultivated fields in Denmark, showed a set of 1541 spiders in total. In the study of Schmidt and Tscharntke [36], the collection of spiders from 26 crop fields and 16 perennial habitats reached in total 4700 specimens, of which 2157 were immature. These authors also showed that 47 species of spiders were found in crop fields. Thorbek and Bilde [5], in their research, showed that spiders were more sensitive to mechanical soil cultivation compared to other arthropods. They also noted that intensive soil cultivation, such as ploughing, did not kill more arthropods than, for example, grass cutting. Our study also did not reveal such a negative role of ploughing on spider abundance. Concerning the species richness, we did not observe differences between the two tillage systems. Rush et al. [11] showed that the agricultural system had no effect on the species richness of spiders. They also indicate that homogeneous landscapes favour only a few species well adapted to arable fields. The most numerous spiders were from the Linyphiidae family (76% of collected specimens), which is able to easily colonise new areas, with the dominant species being *O. apicatus*, *E. atra,* and *E. dentipalpis*. These agrobiontic species, typical of agricultural plantations in Europe, were more numerous in ploughed than in ploughless fields. In addition, Glück and Ingrisch [37] found that total spider abundance was higher in conventional than in no-till plots because of a larger number of Linyphiidae in conventional plots. The dominant species, *O. apicatus,* was captured in the highest number in the ploughless field cropped with triticale. This species is an agrobiont most frequently present in large fields [18,38–40]. One characteristic of agrobiont spiders is the synchronisation of their life cycle with the arable crop growing season; they reach adulthood and reproduce during the main part of the plant growing period [16]. Agrobiont species are small pioneer species that have a suitable dispersal ability and are characteristic of frequently disturbed areas such as grasslands and cereal fields [15,16]. These species are also dominant in arable fields in other European countries [5,16,18,21,39,40].

Although *O. apicatus* builds small webs near the ground, several authors consider it to be an actively hunting cursorial species [41]; however, it is definitely a species adapted to air dispersion, which enables it to make long-distance migratory trips and ensures its quite even distribution in large fields [18,38,42]. Its density in perennial habitats is often lower than in cropped fields [18,41,43]. The eudominant species *E. atra* and *E. dentipalpis* occurred in ploughless and ploughed fields in similar numbers, but *E. dentipalpis* and *B. gracilis* proved to be a species characteristic of ploughless soil tillage, as evidenced by the IndVal indicator. These web-building species consume cereal aphids and catch considerable numbers of them in horizontal sheet webs, which can cover up to half of the surface area of a wheat field [44]. According to Downie et al. [2], *E. dentipalpis* preferred more compacted and established grassland areas, in contrast to *E. atra*, which was common in autumn-sown crops.

Ballooning is a strategy by which spiders can recolonise many agroecosystems [25,45,46]. Aerial dispersal on silk threats allows common agrobiont linyphiids (e.g., *Oedothorax* spp., *Erigone* spp., *T. tenuis*, and *B. gracilis*) to colonise adjacent fields throughout the entire plant growing season but, as in our study, their most intensive activity is observed as air temperatures and ground population densities rise [47]. The source of early-season spider migrations to arable fields are mostly semi-natural habitats, such as balks, grassy margins, and perennial crops, which have been reported by many authors [18,25,41,42,48–50].

One of the answers to the first hypothesis, which was verified to be true only in part, namely lower biodiversity of spiders in fields with ploughed soil, can be found in the

article by Entling et al. [24], where the researchers showed that spiders from disturbed habitats ballooned 5.5 times more frequently than spiders from stable habitats. Hence, a certain amount of disturbance can be observed in ploughed fields, a consequence of which is an altered structure of dominance of the analysed invertebrates, where the share of one species considerably exceeds in its abundance all the other species in a given assemblage. The study indicated that the agrobiont linyphiids' combination of high dispersal abilities and high reproductive rate enables them to exploit the transient resources of different habitats in the agricultural landscape [51]. The distinctly seen increase in the number of spiders noted in July may have been caused by the cutting of oilseed rape or spring cereal fields neighbouring the fields in our study, which was a factor inducing the dispersal of spiders [16]. A genetic study on the feeding habits of an epigeal spider community in a winter wheat agroecosystem, and correlating these results with prey availability, indicate that spiders of the Linyphidae family foraging mostly on Collembola prey also have the potential to play an important role in suppressing early-season aphid populations [22]. The higher abundance of species of the Linyphiidae family in ploughed systems may have been a result of the lesser competition from spiders of the Lycosidae family and the fact that they are potential prey for species from that family [14,52].

In our study, cursorial spiders, mainly those that belong to the Lycosidae, appeared in higher numbers in ploughed fields, especially cropped with wheat, where the species *Pardosa prativaga* was evidently the dominant one. The microclimate in the wheat fields, together with the agrotechnology applied there were probably factors that determined the appearance of spiders with larger body sizes. In a study by Pfiffner and Luka [39], *P. prativaga* inhabited the field centre in high numbers, and the researchers found very uniform spider assemblages in conventional fields. Other dominant species, such as *Pardosa palustris*, occurred in higher numbers in ploughless fields, while *Pardosa paludicola* was more numerous in ploughed fields. Rush et al. [11] also concluded that *P. prativaga* preferred conventionally managed fields, whereas *P. palustris* and *Pardosa agrestis* were more common in organically managed fields. According to Öberg et al. [18], *P. agrestis* and *P. palustris* dominated cereal organic fields and their surroundings in Sweden, but the latter species was positively correlated with a high proportion of perennial crops in the surrounding landscape and showed a preference for the field margin. Pedley and Dolman [53] showed that greater disturbance intensity selected spiders with larger body sizes and cursorial species, both associated with an active hunting strategy. These researchers maintain that recolonisation of this group of spiders takes place via terrestrial dispersal, in contrast to the aerial dispersal of *P. agrestis*, the main cursorial agrobiont spider in Hungary [9,16]. In a study by Birkhofer et al. [54], species of the genera *Pardosa* spp. and *Xysticus* spp. were distinguished by their high consumption of the grain aphid (*S. avenae*) and were able to control the growth of this aphid's population in the early phase of the infestation.

The study by Michałko and Birkchofer [55] indicates that non-crop habitats that are more similar to local crop species are most suitable as a potential source of habitats for agrobiont spiders. They also suggest that both web-building and cursorial species characteristic for cereal fields inhabited mostly meadows among all non-crop habitats. Indeed, the composition of functional traits is more similar between spider assemblages in grasslands and cereal fields than in forests [10]. Using the IndVal method, we determined six significant species of spiders characteristic for ploughed fields, which were qualitatively different from species typical of ploughless fields. Changes taking place in the studied fields in response to conventional soil tillage treatments proved to be more beneficial for the most numerous group of agrobionts. It was found that ploughed soil tillage favours the presence of agrobiontic species, whereas non-inversion tillage increases the biodiversity of spider assemblages by creating an ecologically more stable habitat.

## 5. Conclusions

The type of soil tillage carried out can shape the composition of arachnofauna in the agroecosystems. Conventional ploughing soil tillage can entail considerable disturbances

to the environment, promoting mostly web-building agrobionts and epigeic active hunters. Simplifications in soil tillage, such as ploughless systems, create an opportunity to improve the biodiversity of spiders and to stabilise their assemblages. This is particularly important considering the role that these predatory arthropods play, namely controlling populations of crop pests.

**Author Contributions:** Conceptualisation, investigation, writing E.T.; software, formal analysis A.K.; data curation, visualisation M.N.; taxonomic classification Ł.T.; taxonomic classification Ł.N.; taxonomic classification I.H. All authors have read and agreed to the published version of the manuscript.

**Funding:** The results presented in this paper were obtained as a part of a comprehensive study financed by the University of Warmia and Mazury in Olsztyn, Faculty of Agriculture and Forestry, Department of Entomology, Phytopathology and Molecular Diagnostics. This work was also supported by a research project of the University of Warmia and Mazury in Olsztyn (no. 30.610.010-110).

**Institutional Review Board Statement:** Not applicable.

**Informed Consent Statement:** Not applicable.

**Data Availability Statement:** Data relating to the abundance of studying spiders assemblages presented in this study are available on request from the corresponding author.

**Acknowledgments:** We are grateful to the anonymous reviewer for valuable comments improving our manuscript.

**Conflicts of Interest:** The authors declare no conflict of interest.

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
