# Peer review of "Non-Inversion Tillage as a Chance to Increase the Biodiversity of Ground-Dwelling Spiders in Agroecosystems: Preliminary Results"

_agronomy, doi:10.3390/agronomy11112150_

Round 1
Reviewer 1 Report
This work involves great difficulty due to the large number of specimens captured from a group that is not always easy to determine, taking into account that, on many occasions, they will surely have been immature specimens.
I have to say that the only thing I miss is that the test be repeated another year to really confirm the increase in spider biodiversity with this less invasive soil tillage systems. Even so, the work is well planned.
Review and changes:
Line 36: Keyworks: It would be advisable, although not mandatory, to put a minimum of 5 keywords. Give more visibility to article.
On Line 147 (Table 1) you would delete “(Abreev. - Abbreviation)”, since it’s already implicit in the text. In the same table, it would also be convenient to eliminate the underscores “_” placed between the words in the table header. This is necessary to do the data analysis, but is rare or inconspicuous in the final data output.
On line 177, when it says “(Figure 1c)”, should it actually put (Figure 1b)? and on line 187, when it says "(Figure 1b)", Should it enter (Figure 1c)? In addition, in Figures 1a, 1b and 1c and 3a, 3b and 3c, it would be convenient to change one of the markers (squares or dark circles) to one without filling to better differentiate them. Also, in those same figures, to provide clarity and simplify their reading, on the ordinate axis it would be convenient to leave only the day and month sampled (the year is already known since there is only one).
On line 204 there is a missing space between "& Grüm", and should remain "& Grúm".
On line 201 it is indicated that the group of Superdominant species is composed of 44.25%, however, in table 3, it appears 44.15%.
On line 245 separate and delete the comma "," between the spider species and the author.
Between lines 244-246, I make an observation. Table 1 already specifies species of spider and author, it’s not necessary to put it in the rest of the text, just mentioning it once is enough. But in case of putting it, either the author is put in all or not in any.
In Table 4, need to align the titles “Non-investment” and “Plought”.
Author Response
Our answer is in the attached file (word) and marked as an answer for Reviewer 1

Reviewer 2 Report
The ms presents the results of the study on assemblages of Araneae in large plantations of triticale, wheat and barley, cultivated under two different systems of soil tillage: conventional with ploughing and non-inversion tillage, in north-eastern Poland. The authors put hypothesis that ploughing causes a decrease in the abundance and species richness of spiders; in fields under the non-inversion tillage system, more active epigeic hunters (cursorials) spiders can be found whereas web spiders are more abundant in conventional fields; in non-inversion fields, the activity of spiders is even throughout the whole plant growing season.
Study design to check the hypothesis seems adequate, as well as applied statistical analyses. I have found the ms written very well, with a good structure, though have some concerns regarding English, but since I am not a native English speaker, I suggest to authors checking the language. Discussion could be a bit more structured, it is a bit harder to follow than the rest of the ms.
The work seems relevant and reveals some aspects of studied managements that promote spiders diversity in the field, like species characteristic for ploughless soil tillage E. dentipalpi and B. gracilis, an important predators on cereal aphids. Regarding functional diversity, conventional ploughing soil tillage in the studied agroecosystems promoted mostly web-building agrobionts and epigeic active hunters. After some minor corrections and clarifying the discussion, the ms is worth of publishing.
Comments throughout the ms:
17 remove „ Significantly“
26-32 could it be rearranged and simplified? e.g. „According to IndVal, Erigone dentipalpis (Wider) and Bathyphantes gracilis (Black-29 wall), were significantly characteristic (p<0.05) for non-inversion soil tillage, whereas six species, Oedothorax apicatus (Blackwall), Pardosa prativaga (L. Koch), Pardosa paludicola (Clerck) Pachygnatha clerki Sundevall, Dicimbium nigrum brevisetosum Locket and Clubiona reclusa O.P.-Camblidge were typical of the soil tillage with ploughing. “
41 „ they give preference“ Who? EU? Maybe just to put „with preference“ or put on a passive form „The preference have been given to ….“
55 Spiders, in addition, to listed hunting strategies are also ambushing hunters, species of Gnaphosidae family (Michakej et al. 2019, Scientific Reports)
73 What treatments besides ni are considered “Simplified soil tillage systems”? - so maybe to expand the sentence “Simplified soil tillage systems such as non-inversion (ni) and ……” You cited references 2 and 22, what simplified systems where studied there? You can list few of them…
87 How often tillage was done? And at what dates. It could be interesting to see the ploughing dates pointed on the Figs 1 and 3.
92 Which different cultivation aggregates were used to loosen and mix soil without inverting it?
94 What are the dimensions (or, capture capacity) of pitfall traps?
94 Why other method of specimen collection were not included? Using vacuum…Maybe trapping web builders and foliage dwelling spiders using D-Vac method (Bali et al 2019 https://doi.org/10.30963/aramit5808)?
95 Not consistent with Table 1
102 Have all spiders been identified to species level? State identification process in more details
103 Suggestion: interesting would be to see if there is a difference in abundance and presence of nocturnal or daily active species in pitfall traps between different treatments
111 Does the data set includes juvenile spiders not identified to species level and how they were treated in the calculation of these indices
115 How sample period was classified, by month, exact date of collection ecc. Maybe to add more details in Material and methods section after the sentence ending in line 98. That in total it makes eight collection dates when traps were emptied.
147 Why there is no B_pl and W_ni?
185 Unclear? What are “spiders found” and what “captured species”?
192 “The most separate group were spiders in the ploughed fields of wheat.” Unclear, can you rephrase it.
Figure 1 C- unclear Can you try with combination of other symbols, nor circle and squares. Need to be enlarged to distinguish at Fig 1c.
Figure 2 should you add that graph presents data for two-dimensional solution, and values of stress=xy (R 2 =xy).
263 What is considered an abundant assemblage? Any reference!
265 What is an assumption why are these species more numerous in this type of plantation. For example, easier colonisation …
298 and 349 – what lower increase in biodiversity means?
349 Species richness –no significant difference, how then increase in biodiversity
Table 1. Maybe consider mowing it to supplementary. Maybe different organisation of the table should be more convenient, alphabetical order maybe changed with more abundant species on top of the table
Table 2. Should type of crop be included in this analysis
Author Response
Our answer is in the attached file (word) and marked as an answer for Reviewer 2

Reviewer 3 Report
General remarks
The work is well theoretically prepared, carefully written, which is not common at this stage of publishing.
The authors taken the problem of the presence of spiders in arable fields cultivated in two different cultivation systems. In my opinion, they undertook a difficult task, because spiders are a very diverse group (as they write themselves). However, it should be clearly shown that it was more about ground-dwelling spiders than specimens on plants, therefore the suggestion of changes in the title, text and table 1. The weakness of the article is relatively the short sampling period, although the collected material is large and allows write conclusions. For this reason, I propose to add (apart from remark L2) to the title "preliminary results"
Another important thing to correctly interpret the results is the methodological data, which is somewhat poor. Both the part concerning the research fields and the catching of spiders are insufficiently described, much more details should be provided as it looked in practice. We do not know what the simplifications were, what machines were used, what agrotechnical treatments were performed, both related to soil cultivation and fertilization and plant protection. We do not know the dates of performing agrotechnical treatments, so it is difficult to draw conclusions as to the number and activity of spiders. Interesting results were obtained in this research, which may fill the research gap, but it is necessary to complete the data. Certainly, the numbers of caught spiders as well as the statistical analysis are sufficient here.
The advantage of the article is also the recognition of spiders’ species. Since the species are known, it is also possible to mark, for example, in Table 1, which of them can be classified as epigeic, and which can be classified as the web-aerial group or present on plants (at least what is known).
I also have some questions that may contribute to the discussion of the results obtained. Why are there fewer spiders in the conventional system, do they overwinter in this field, that the treatments are killing them, or maybe they migrate from uncultivated places? Maybe it is worthwhile to enrich the work with this topic, i.e. the role of field margins, mixtures of flowering plants etc. on the occurrence of epigeic arthropods, to state how other epigeic assemblages react. Recently, there has been published some work on this topic on
ground-dwelling spider assemblages, rove beetles or ground beetles as well.
What about web spiders? Can they also be compared here if the method used to catch them was not correct? If it turns out that their numbers were not large, it can be concluded that they are rather in barber's traps by accident.
Are collected species the real natural enemies of cereal pests?
Specific comments
L 2 there is need add “ground-dwelling spiders” to the title instead of arachnofauna. It is a very diverse group of arthropods and pitfall traps used in the experiment can be used to the specific ground-dwelling spiders only
L 49 there is need provide the citations of the statement “Conservation tillage is predominantly ploughless cultivation and typically does not displace soil-dwelling organisms”
L 81 why, whether spiders are then in the field, or do not migrate from the margins, after all, nothing grows in this field then
L85 this hypothesis cannot be substantiated because no methods were selected to test it and the experiment was not planned in a sufficiently long time
L 89-93 be more precise, there is not too much information how the researched fields looks like. Fields 80-90 ha but which one was 60, which 80? What grew then maybe it doesn't really important, but writing about triticale and wheat or triticale and barley sounds like I have to choose a crop here
L 93 The authors automatically proceed to the method of collected spiders, without separating it with any paragraph, and this is the material that should be found elsewhere in the methodology
L 91 what cultivars of triticale and wheat/barley were cultivated, say more if it is known?
L 93-94 move to the chapter Data collection
L 95 how the traps were arranged, in line? Where exactly in studied field?
L 143 add information about the preferences of individual species (it was mentioned earlier in the text)
L 204 it is not necessary to put percentages up to two decimal places, both in table 3 and in the text
Author Response
Our answer is in the attached file (word) and marked as an answer for Reviewer 3
